# Exposure of patients to di(2-ethylhexy) phthalate (DEHP) and its metabolite MEHP during extracorporeal membrane oxygenation (ECMO) therapy

Franziska Kaestner[1¤a], Frederik Seiler[1], Daniel Rapp[2¤b], Elisabeth Eckert[3], Johannes Müller[3], Carlos Metz[1], Robert Bals[1], Hans Drexler[3], Philipp M. Lepper[1]*, Thomas Göen[3]

**1** Department of Internal Medicine V, University Hospital of Saarland, Homburg, Germany, **2** Institute for Medical Biometry, Epidemiology and Medical Informatics, Saarland University, Homburg/Saar, Germany, **3** Institute and Outpatient Clinic of Occupational, Social and Environmental Medicine, Friedrich-Alexander-Universität Erlangen-Nürnberg, Erlangen, Germany

¤a Current address: Department of Respiratory Medicine, Fachkliniken Wangen, Wangen, Germany
¤b Current address: Department of Neurology, RKU University and Rehabilitation Clinics Ulm, Ulm, Germany
* philipp.lepper@uks.eu

## Abstract

The plasticizer di(2-ethylhexyl)phthalate (DEHP) is often used for PVC medical devices, that are also largely used for intensive care medical treatments, like extracorporeal membrane oxygenation (ECMO) therapy. Due to the toxicological potential of DEHP, the inner exposure of patients with this plasticizer is a strong matter of concern as many studies have shown a high leaching potential of DEHP into blood. In this study, the inner DEHP exposure of patients undergoing ECMO treatment was investigated. The determined DEHP blood levels of ECMO patients and the patients of the control group ranged from 31.5 to 1009 μg/L (median 156.0 μg/L) and from 19.4 to 75.3 μg/L (median 36.4 μg/L), respectively. MEHP blood levels were determined to range from < LOD to 475 μg/L (median 15.9 μg/L) in ECMO patients and from < LOD to 9.9 μg/L (median 3.7 μg/L) in the control group patients, respectively. Increased DEHP exposure was associated with the number of cannulas and membranes of the ECMO setting, whereas residual diuresis decreased the exposure. Due to the suspected toxicological potential of DEHP, its use in medical devices should be further investigated, in particular for ICU patients with long-term exposure to PVC, like in ECMO therapy.

## Introduction

Modules of polyvinylchloride (PVC) are frequently used as components of medical devices [1]. PVC material contains a high share of plasticizers, which are not chemical bound to the PVC itself and can thus easily migrate into contact liquids [1, 2]. For the most prominent plasticizer di(2-ethylhexyl)phthalate (DEHP) a high leachability from medical devices, like blood

**Data Availability Statement:** All relevant data are within the manuscript and its supporting Information files.

**Funding:** The authors received no specific funding for this work.

**Competing interests:** FK received travelling fees from Getinge Deutschland GmbH; PML received travelling fees from Getinge Deutschland GmbH, and speaker fees from Mitsubishi Pharma, and CSL Behring. The other authors declared no conflict of interest. This does not alter our adherence to PLOS ONE policies on sharing data and materials.

bags and transfusion lines, into blood has been demonstrated [3, 4, 5]. This may result in a considerable internal exposure of the patients. This is a strong cause of concern, as DEHP is potentially linked with several toxicological effects on humans. In particular, the suspected adverse developmental and reproductive effects of DEHP are a constant matter of concern [6, 7]. Additionally, DEHP in much higher values is classified as a carcinogen for which a non-genotoxic mode of action is predominant [8]. The application of several phthalates is regulated in the European Union and DEHP and several other phthalates are banned for the manufacturing of children's toys [9, 10]. In contrast, a restriction of DEHP use in medical devices does not exist. Nevertheless, the European Scientific Committee on Emerging and Newly-Identified Health Risks recommended a substitution of DEHP for medical devices, whenever possible [11]

Studies in neonatology reported increased bilirubin levels in patients associated with DEHP exposure from nutrition lines [12]. Elevated bilirubin levels were also observed in patients undergoing extracorporeal membrane oxygenation (ECMO) treatment [13, 14]. Reasons for increasing bilirubin levels in Intensive Care Unit (ICU) patients can be, among others, ischemic, toxic, septic or due to hemolysis [15]. Elevated bilirubin can be a lead for early liver dysfunction which is known for increased morbidity and mortality [16]. Toxic reasons are mostly seen due to medication [17]. Only few studies dealt with the effects of DEHP exposure in ICU patients [18, 19, 20]. Thus, the purpose of this study was the assessment of DEHP and MEHP levels in blood and MEHP metabolites in urine of ECMO patients.

## Materials and methods

### Study design

For this explorative cross-sectional study, routinely collected data from patients were used who were hospitalised between May 2015 and December 2016 on a pneumology focused intensive care unit (ICU) of the Saarland University Medical Center, Germany. Additionally, blood and urine samples were collected once during ECMO treatment in order to determine the levels of DEHP and its metabolite. Urine samples were only taken of patients with intact urine excretion. In four patients, DEHP levels were additionally determined prior to the initiation of ECMO and also once during the course of treatment (seven days after the start of ECMO treatment) in order to perform a descriptive pre-post analysis. Verbal informed consent was obtained from participants or legal representatives; no minors were included in the study. The study was approved by the ethics committee of the Medical Association of the Federal State of Saarland (Ärztekammer des Saarlandes). Data were shared and explored in accordance with patient consent.

### Study population

The study collective included 17 patients receiving ECMO treatment, of which seven were female and ten were male. 13 patients died during ECMO treatment. The age of the patients ranged from 20 to 72 years (median 51 years).

The main cause for the initiation of ECMO therapy was the development of an acute respiratory distress syndrome (ARDS). Three patients were suffering from cystic fibrosis, five patients had lung fibrosis, three suffered from right heart decompensation due to pulmonary hypertension and one due to embolism. Two had ARDS because of pneumonia and one because of aspiration. The simplified acute physiology score (SAPS) as a measurement for severity of an illness had a mean of 39 points at the time of sample collection. The oxygenation index (pO2/FiO2) as an index of lung function on ventilator ranged from 54 to 154 (median 77) prior to ECMO treatment. ECMO therapy was started when the oxygenation deteriorated

persistently with an oxygenation index below 100 or a blood pH value of less than 7.2 despite other procedures.

Run time of ECMO ranged from two to 65 days (median 31 days). The sample collection for the assessment of internal DEHP exposure took place after a median ECMO run time of 11 days (range 2 to 54 days). Priming of the ECMO circuit was done with physiological saline solution. The percutaneous cannulation was controlled by ultrasound, guided by a team of trained specialists in intensive care medicine. All study patients received high-flow ECMOs with a 7 L oxygenator (CardioHelp, Maquet, Rastatt, Germany). To prevent clotting, patients were treated with heparine (iv) or agatroban in case of a known or developing heparine-caused thrombocytopenia. Sedation was administered according to a protocol.

14 patients were cannulated veno-venous (vv), three veno-arterial (va). Standard vv cannulation was femoral drainage (23 F) and jugular return (19 F). Four patients had to be changed from vv to either vva with a venous drainage and one venous return and one arterial return or vvv with two draining cannulas. Five patients needed one change of membrane because of clotting, one needed four changes. In nine patients two cannulas were utilized, six patients who needed three cannulas in order to maximize the flow or because of clotting, one patient got in total four cannulas and one patient five.

In total, nine patients needed hemofiltration due to renal failure, of whom five had no urine excretion anymore. In one patient, a cytokine adsorber (CytoSorbents Corporation, New Jersey, USA) was used.

Blood products were administered according to guidelines and in case of severe bleeding. In brief, blood was transfused if Hb was below 7.0 g/dL or if central venous oxygenation was below 65% despite adequate cardiac output. The amount of given packed red blood cells ranged from two to 53 bags á 250 mL each (median 12 bags).

Additionally, a control group of five patients (two male, three female) was included. The age ranged from 49 to 74 years (median 59 years). The control group patients were also ICU patients who were on a ventilator for five to 17 days (median 11 days) until sample collection, but received no ECMO treatment. The SAPS score at the time of sample collection was 30 to 54 (median 41). One was a dialysis patient but still had urine excretion. One of the five control group patients died while on ventilation.

For details about the study population and its treatment see **Table 1**.

## Analysis of blood and urine samples

The quantification of DEHP and MEHP in the blood samples was carried out according to a previously published procedure [2]. In brief, the analytes were extracted from the blood using liquid-liquid extraction, separated using liquid chromatography, and detected by tandem mass spectrometry. The limits of quantification (LOQ) were 5.0 μg/L for DEHP and 2.0 μg/L for the metabolite MEHP. An analysis of spiked quality control samples (QC) indicated a precision range of 4.4–5.3% and an accuracy (recovery) of 100–109% for both analytes.

The urine samples were processed according to a previously published procedure [21] that enables the determination of the primary metabolite MEHP and three secondary DEHP metabolites, namely mono-(2-ethyl-5-hydroxyhexyl) phthalate (5OH-MEHP), mono-(2-ethyl-5-oxohexyl) phthalate (5oxo-MEHP) and mono-(2-ethyl-5-carboxypentyl) phthalate (5cx-MEPP) in urine. Briefly, the urine samples were initially enzymatically hydrolyzed. The sample cleanup and enrichment of the analytes was accomplished by the application of an online cleanup procedure using a restricted access material. Subsequently, the analytes were chromatographically separated on an analytical column, and detected using tandem mass spectrometry. The limit of quantification was 0.5 μg/L for each analyte. An analysis of spiked

**Table 1. Study group and control group description.** *: at time of DEHP measurement (ECMO = extracorporeal membrane oxygenation, SAPS = simplified acute physiology score, DEHP = di(2-ethylhexyl)phthalate, MEHP = mono(2-ethylhexyl)phthalate, PC = packed cells, CF = cystic fibrosis, ILD = interstitial lung disease, PH = pulmonary hypertension, PE = pulmonary embolism, ARDS = acute respiratory distress syndrome, CLAD = chronic lung allograft dysfunction, AECOPD = acute exacerbation of chronic obstructive pulmonary disease).

| | No ECMO (n = 5) | ECMO (n = 17) | P value |
|---|---|---|---|
| Age (years) | 59.0 (49.0;74.0) | 51.0 (20.0;72.0) | 0.055 |
| Time on respirator (days) | 11.0 (5.00;17.0) | . (.;.) | . |
| Time on ECMO (days) | . (.;.) | 31.0 (2.00;65.0) | . |
| Horowitz index | . (.;.) | 77.0 (55.0;154) | . |
| SAPS score | 41.0 (30.0;54.0) | 39.0 (13.0;72.0) | 0.666 |
| Time to DEHP monitoring (days) | 10.0 (5.00;13.0) | 11.0 (2.00;54.0) | 0.408 |
| Bilirubin * (mg/dL) | 0.40 (0.30;1.00) | 1.30 (0.20;19.4) | 0.091 |
| Bilirubin max (mg/dL) | 0.60 (0.40;2.20) | 4.60 (0.20;24.0) | 0.055 |
| Lactat * (mmol/L) | 1.40 (1.00;2.40) | 1.00 (0.40;16.0) | 0.382 |
| ECMO flow * (L/min) | . (.;.) | 3.55 (1.00;5.20) | . |
| ECMO flow max (L/min) | . (.;.) | 5.00 (4.00;7.00) | . |
| DEHP (μg/L) | 36.4 (19.4;75.3) | 156 (31.5;1009) | 0.007 |
| MEHP (μg/L) | 3.67 (0.00;9.94) | 15.9 (0.00;475) | 0.075 |
| PC before DEHP (units) | 2.00 (0.00;14.0) | 12.0 (2.00;53.0) | 0.030 |
| DEHP Equivalent (nmol/L) | 110 (75.2;193) | 496 (80.6;3601) | 0.007 |
| MEHP share (%) | 9.30 (0.00;34.0) | 19.6 (0.00;54.1) | 0.782 |
| Diagnosis: | | | 0.187 |
| CF | 0 (0.00%) | 3 (17.6%) | |
| ILD | 0 (0.00%) | 5 (29.4%) | |
| Right heart decompensationPH | 0 (0.00%) | 3 (17.6%) | |
| Right heart decompensation PE | 0 (0.00%) | 1 (5.88%) | |
| ARDS pneumonia | 3 (60.0%) | 2 (11.8%) | |
| ARDS aspiration | 1 (20.0%) | 1 (5.88%) | |
| Others (CLAD, AECOPD) | 1 (20.0%) | 2 (11.8%) | |
| Death: | | | 0.039 |
| 0 | 4 (80.0%) | 4 (23.5%) | |
| 1 | 1 (20.0%) | 13 (76.5%) | |
| Membranes: | | | . |
| 1 | 0 (.%) | 10 (62.5%) | |
| 2 | 0 (.%) | 5 (31.2%) | |
| 5 | 0 (.%) | 1 (6.25%) | |
| Cannulas: | | | . |
| 2 | 0 (.%) | 9 (52.9%) | |
| 3 | 0 (.%) | 6 (35.3%) | |
| 4 | 0 (.%) | 1 (5.88%) | |
| 5 | 0 (.%) | 1 (5.88%) | |
| Hemofiltration: | | | 0.323 |
| 0 | 4 (80.0%) | 8 (47.1%) | |
| 1 | 1 (20.0%) | 9 (52.9%) | |
| Urine output: | | | 0.290 |
| 0 | 0 (0.00%) | 5 (29.4%) | |
| 1 | 5 (100%) | 12 (70.6%) | |

quality control samples (QC) indicated a precision range of 7.3–11.0% and an accuracy (recovery) of 92–105% for all analytes. Moreover, the proficiency of the procedure was proved by the successful participation in the proficiency test program of the German External Quality Assessment Scheme (G-EQUAS; [22]).

### Statistical analysis

The statistical analyses were performed using R software version 3.5.0 (R Core Team (2016): R: A language and Environment for Statistical Computing, Vienna Austria, URL: www.r-project. org). Continuous characteristics were described as median [Minimum; Maximum] and categorical characteristics as n (%). Comparison of two independent groups was performed using Mann-Whitney U test for continuous characteristics and Fisher's exact test for binary characteristics. No statistical tests were performed to compare categorical characteristics with more than two categories because of the limited number of cases. Statistical significance was assumed for p-values of less than 0.05 using a two-sided significance level of $\alpha = 5\%$. There was no correction of p-values in this explorative analysis.

## Results

The determined DEHP blood levels of the ECMO patients and the patients of the control group ranged from 31.5 to 1009 µg/L (median 156.0 µg/L) and from 19.4 to 75.3 µg/L (median 36.4 µg/L), respectively. MEHP blood levels were determined to range from < LOD to 475 µg/ L (median 15.9 µg/L) in the ECMO patients and from < LOD to 9.9 µg/L (median 3.7 µg/L) in the control group patients, respectively (Fig 1).

In four patients, DEHP and MEHP blood levels were also determined prior to ECMO initiation and compared to the respective levels after an ECMO runtime of seven days. In each but one patient, this resulted in significantly elevated blood levels of DEHP and MEHP (Fig 2) with determined median levels of 40.7 µg DEHP/L and 9.5 µg MEHP/L prior to ECMO treatment compared to 74.4 µg DEHP/L and 18.3 µg MEHP/L after seven days of ECMO runtime. The decrease of MEHP after seven days of ECMO in one patient may be due to a limited metabolism but is not verified.

The association between duration of ECMO therapy and DEHP or MEHP was tested using Spearman's correlation coefficient. However, the correlation did not reach statistical significance (r = 0.16, P = 0.53).

Median blood level of DEHP and MEHP in patients with only two cannulas was 156 µg/L and 9.5 µg/L. Median blood level of DEHP and MEHP in patients with more than two cannulas was 338 µg/L and 114.9 µg/L. Patients with only one membrane had a median DEHP level of 136 µg/L and MEHP of 17 µg/L. Patients that needed more than one membrane had a median DEHP of 535 µg/L and MEHP of 114.9 µg/L (Figs 3 and 4).

Correlation between DEHP and applied packed cells showed significance (spearman's correlation coefficient r = 0.59, p<0.01).

Patients without urine output showed significantly increased DEHP and MEHP blood levels (median 823 µg/L and 214 µg/L) in comparison to the other patients (median 75.3 µg/L and 7.11 µg/L) (Fig 5).

In one patient DEHP and MEHP blood levels were determined before (median DEHP 256 µg/L and MEHP 19 µg/L) and after dialysis (median DEHP 289 µg/L and MEHP 24 µg/L).

In the collected urine samples the main DEHP metabolites 5oxo-MEHP, 5OH-MEHP and 5cx-MEPP were determined. 5oxo-MEHP ranged from 2.82 µg/L to 2066.3 µg/L (median 522 µg/L), 5OH-MEHP 6.84 µg/L to 7402 µg/L (median 3359 µg/L) and 5cx-MEPP 12.95 to 24073 µg/L (median 5375 µg/L) (Table 2).

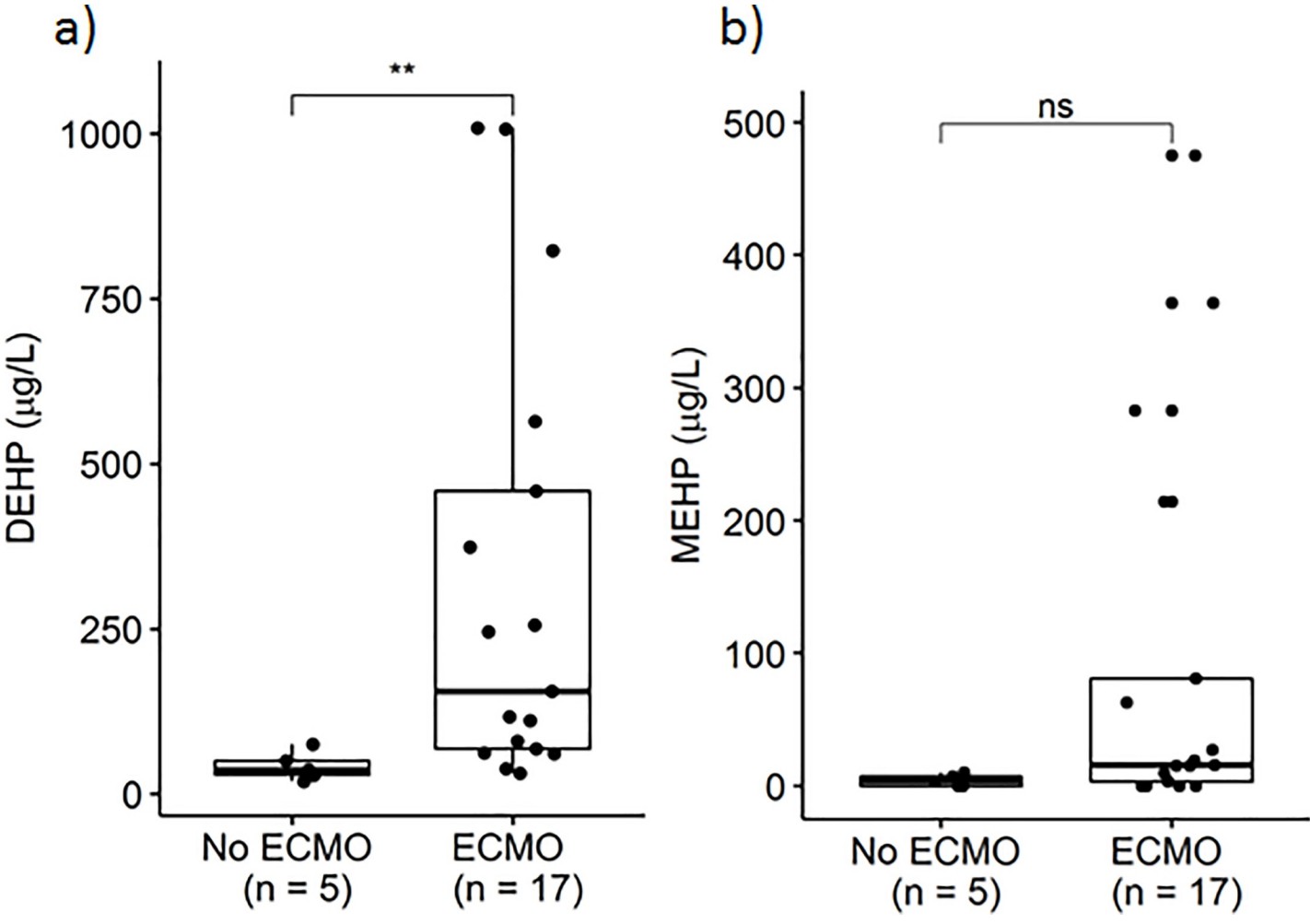

**Fig 1.** Blood levels of DEHP (a) and MEHP (b) in patients without ECMO therapy (control group) and with ECMO therapy (both p = 0.005).

## Discussion

In the present study, we found considerable amounts of the plasticizer di(2-ethylhexy)phthalate (DEHP) and its metabolite MEHP in ECMO patients. Runtime and amount of ECMO cannulas seemed to influence DEHP and MEHP levels.

ECMO is a special extracorporeal medical therapy that provides prolonged cardiac and respiratory support to patients. However, its application generally requires the extensive use of plasticized PVC components, bags and tubes with long lasting blood contact. Because of its characteristics, DEHP is often favored as an additive in blood bags and ECMO devices despite the fact that DEHP migrates very easily from PVC material into blood [4, 23, 24], which may result in a significant inner DEHP exposure of the patients. For healthy individuals, urinanalysis of DEHP metabolites is the established exposure monitoring strategy [21, 25]. ECMO patients, however often lack a residual diuresis which disadvantages the urinanalysis. Thus, for the monitoring of DEHP exposure, analysis in EDTA blood samples of the patients was carried out primarily.

There are already some studies dealing with the question of DEHP migration from PVC medical devices. These studies dealt with the influence of storage time of medical devices [7], of priming procedures [26] and of the applied coating [24, 27]. In the present study high values

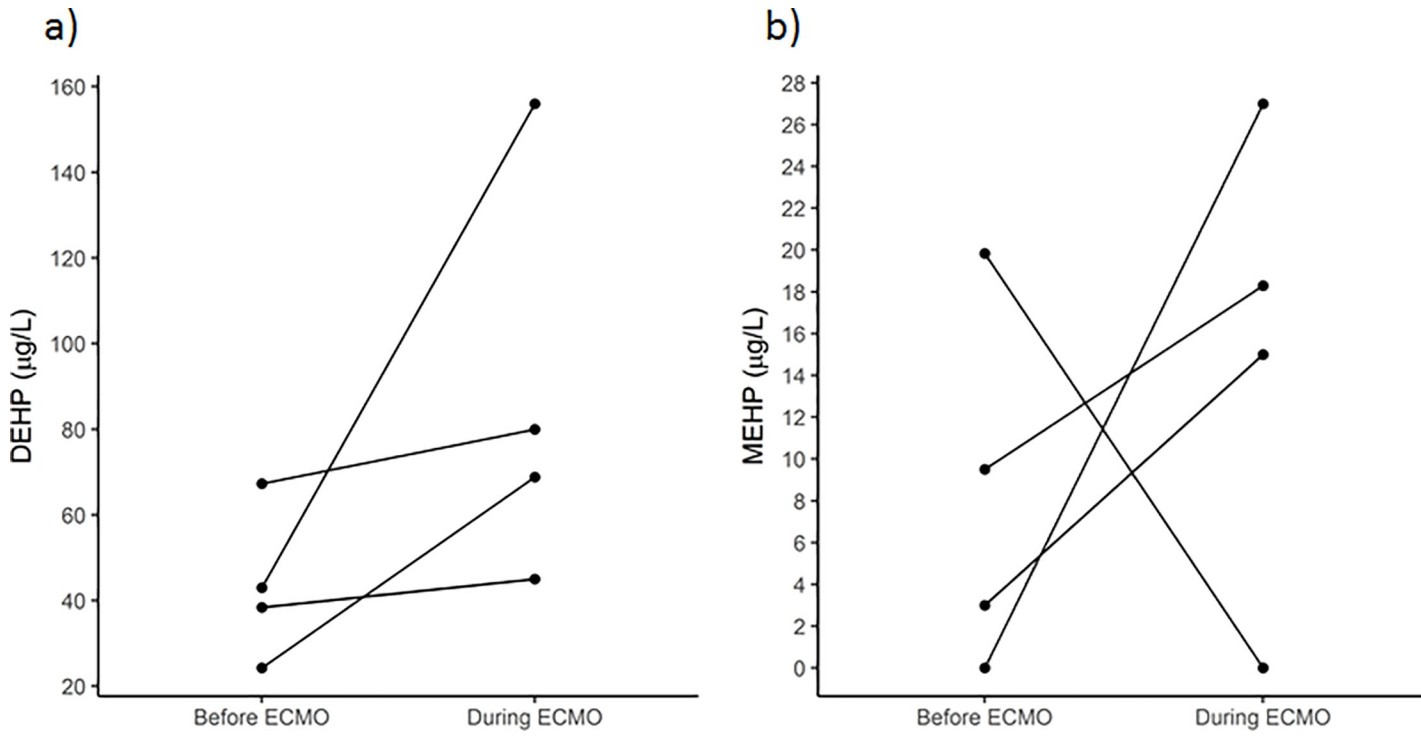

**Fig 2.** Blood levels of DEHP (a) and MEHP (b) in four patients prior to ECMO initiation and after seven days of ECMO therapy.

of DEHP and its metabolites are found to be correlated with longer and more intense ECMO (several cannulas, membranes), but not in all patients. We observed a trend of increased DEHP levels after initiation of ECMO therapy. Furthermore, we observed that DEHP and its metabolites were increased when a more intense ECMO treatment (as measured by number of

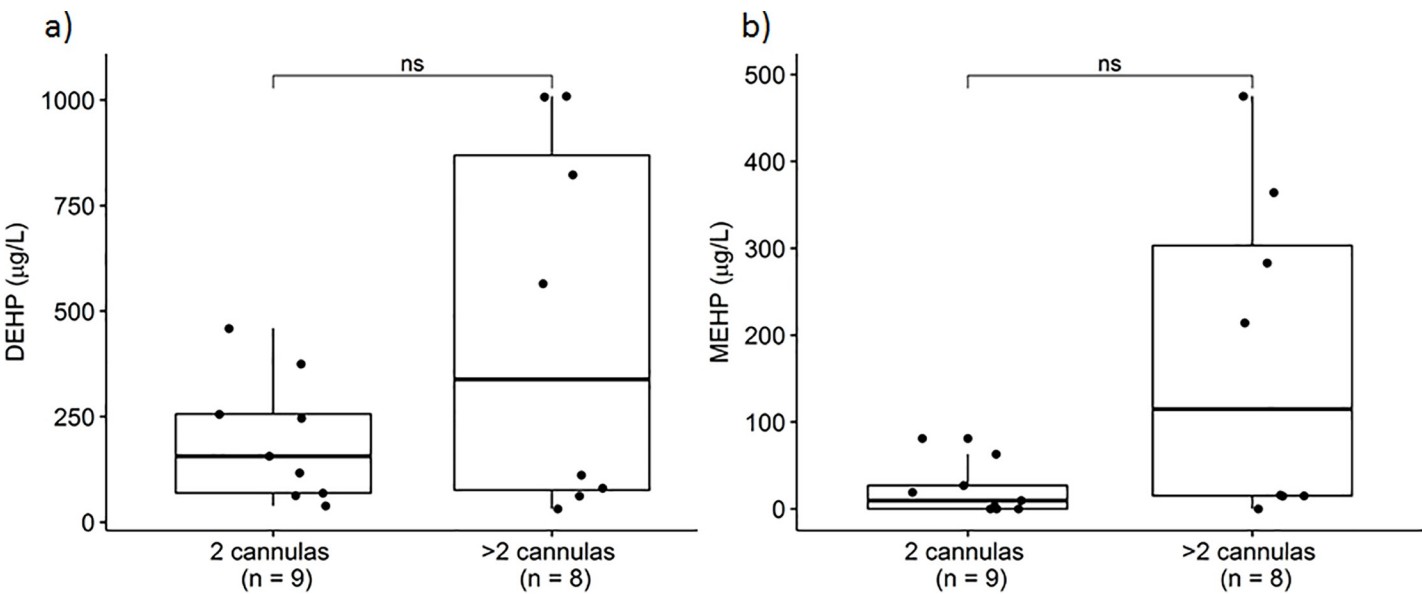

**Fig 3.** Blood levels of DEHP (a) and MEHP (b) in patients with two cannulas and in patients with more than two cannulas (p = 0.541 and p = 0.121, respectively).

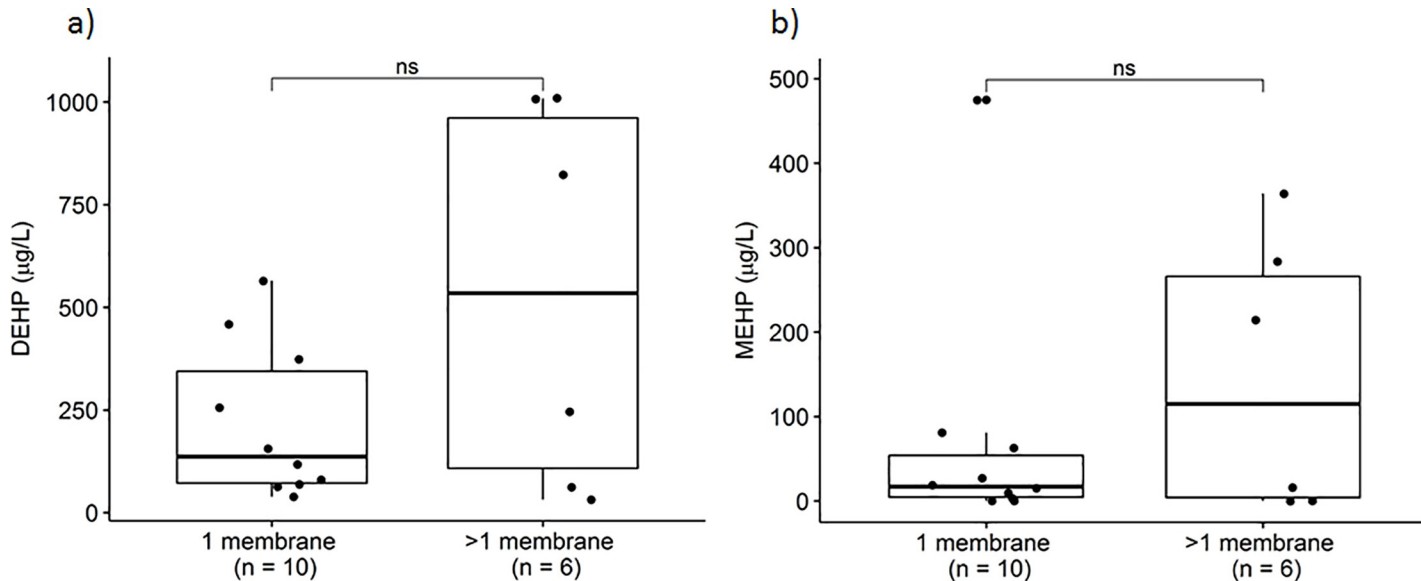

**Fig 4.** Blood levels of DEHP (a) and MEHP (b) in patients with one membrane and in patients with more than one membrane (p = 0.492 and p = 0.702, respectively).

cannulas or number of membranes) was applied. However, there was no positive correlation between duration of ECMO therapy and DEHP levels. This might be due to a rapid increase of DEHP after initiation of the therapy and a steady state during the rest of the treatment. Other factors like the individual metabolism might also have a strong impact on the DEHP-levels. More studies with a larger sample sizes and a more frequent measurement of DEHP-levels are needed in order to further analyse this question.

Accumulation of DEHP in dialysis patients and ICU patients is already known [28, 29]. To our knowledge there are only few studies that dealt with the patients DEHP exposure in relation to the PVC components, like nutrition lines, iv-lines, endotracheal tubes [1, 18] and none

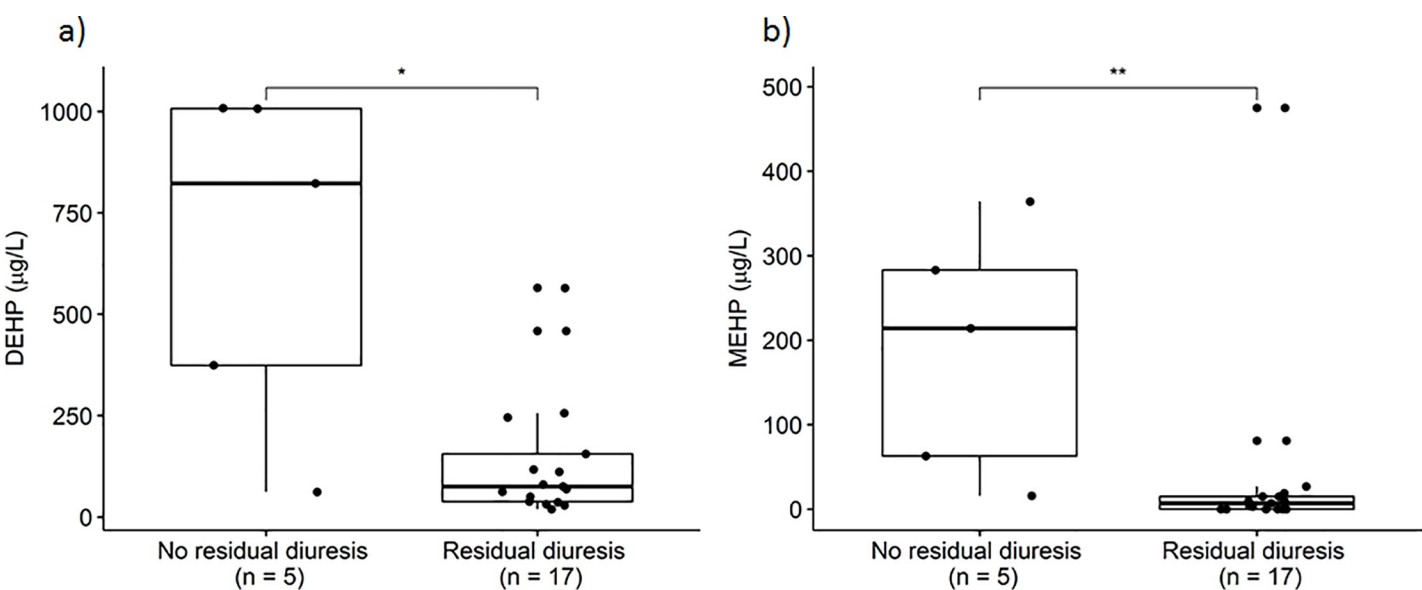

**Fig 5.** DEHP (a) and MEHP (b) in EDTA blood in patients with no residual diuresis and with residual diuresis (p = 0.019 and p = 0.009).

of these in reference to the amount of PVC components during ECMO treatment. Patients undergoing ECMO treatment are known to be prone to numerous complications, including those due to blood clotting or due to other material problems that necessitate an adjustment of the treatment, e.g. additional settings of cannulas or additional filters in the hemofiltration system [30, 31, 32].

We observed a trend towards lower blood levels of DEHP and MEHP in patients with only two cannulas than in patients with more cannulas. Furthermore, patients with only one membrane showed somewhat lower blood levels of DEHP and MEHP than patients that needed more than one membrane. However, probably due to the small sample size, the difference between both groups did not reach statistical significance (**Figs 3 and 4**).

It is known that absorbed DEHP is rapidly metabolized in the body and excreted via urine [8, 25, 33]. It is evident, that ECMO patients with a present urine excretion have a better survival rate than patients without urine output [31, 34]. **Fig 5** shows the determined blood levels of DEHP and MEHP in these patient groups of the here described study. It was observed, that patients without urine output showed significantly increased DEHP blood levels in comparison to the other patients. One possible explanation is the more extensive and longer use of medical devices in patients without active urine output. This is in line with the considerations of the working group of Huygh, who also considered the higher amount of inner DEHP exposure in ICU patients as a result of the increased use of medical devices [18]. The potentially more reasonable explanation for that observation is the inhibited elimination process in patients without regular urine excretion. Clearance of DEHP by dialysis is not described and because of its molecular nature also not to be expected [28, 29, 35]. In the present study, DEHP and MEHP blood levels were determined in one patient before and after dialysis but rather consistent levels were observed. In contrast, a considerable reduction of the DEHP and MEHP contamination in blood products was achieved in a recent study by cell saver treatment including a washing procedure [36]. However, this prevention strategy was not applied during the ECMO treatment of our patients.

We observed a statistically significant correlation between DEHP-levels and number of packed cells given. This could be due to the critical condition of patients who received a high number of packed cells. Those critical patients might also have received a more intense ECMO therapy. On the other hand, we did not measure the DEHP-levels in the blood products. Former studies indicated a high contamination of the blood products with DEHP, which may have contributed to the DEHP-blood levels in those patients [5, 36].

In this study, urine samples were only collected of six patients in the study group and none in the control group. In the study group not all patients had urine excretion and in the control group it was an organizational problem to gain urine samples.

In the collected urine samples the main DEHP metabolites 5oxo-MEHP, 5OH-MEHP and 5cx-MEPP were determined (**Table 2**). Considering the reference values with 20 µg/l for 5oxo-

**Table 2. DEHP metabolites in urine samples and DEHP and MEHP in EDTA blood of six patients.** 5-OH-MEHP = mono(2-ethyl-5-hydroxyhexyl)phthalate; 5-oxo-MEHP = mono(2-ethyl-5-oxo-hexyl)-phthalate; 5-cx-MEPP = mono(2-ethyl-5-carboxypentyl)phthalate.

| Patient | 5OH-MEHP (µg/l) Urine | 5oxo-MEHP (µg/l) Urine | 5cx-MEPP (µg/l) Urine | DEHP (µg/l) EDTA-Blood | MEHP (µg/l) EDTA-Blood |
|---|---|---|---|---|---|
| 1 | 6.84 | 2.82 | 13.0 | 117 | 3 |
| 2 | 6020 | 1720 | 10800 | 459 | 81 |
| 3 | 697 | 221 | 36.1 | 256 | 19 |
| 4 | 7400 | 2070 | 24100 | 565 | 475 |
| 5 | 6680 | 618 | 9650 | 156 | 27 |
| 6 | 405 | 425 | 1100 | 80 | 15 |

MEHP und 30 μg/l for 5OH-MEHP and 5cx-MEHP it is in evidence that 5 of 6 patients have values wide above the reference values determined by the biomonitoring commission [37]. The reason for patients 1 low values stays indistinct. The characteristics of this one patient was five days of ECMO run time with no change in system, only four bags of packed cells given prior to testing DEHP and its metabolites and bilirubin and lactate not elevated at that time.

The determined MEHP proportion in the blood samples varied considerably. In ECMO patients with residual diuresis MEHP proportion was in median 9.44% and in non-residual diuresis patients in median 26.5%. The results support the hypothesis that the inhibited renal clearance may affect the accumulation of DEHP and particularly of the hydrophilic DEHP metabolites in blood. It can also be assumed, that the patients medication may also affect metabolism of DEHP and this causes varying results of measured DEHP. Additionally excretion by sweat also seems to be an excretion pathway for phthalates [38]. The perspiration of intensive care patients differs strongly due to possible high fever and administered drugs and may also contribute to the observed differing results. DEHP and MEHP showed similar dependence, but the MEHP to DEHP ratio was not constant.

Potential adverse health effects of elevated DEHP exposure in patients are widely discussed [7, 18, 27, 35, 39]. Elevated bilirubin levels point to liver dysfunction, which is associated with higher morbidity and mortality [16]. A study on a neonatal ICU observed elevated bilirubin levels in several patients and DEHP exposure due to the used nutrition lines was discussed as a probable cause. Following an equipment replacement to DEHP-free nutrition lines, a significant decrease in bilirubin levels was observed [12]. Other studies also observed elevated bilirubin levels in ECMO patients but did not consider a probable DEHP exposure [34, 40]. Pappalardo even used this parameter in risk scores [41].

In this study, bilirubin levels were determined at the time of sample collection for the assessment of inner DEHP exposure. Generally, ECMO patients exhibited higher bilirubin levels with a median of 1.3 mg/dl (range 0.2–19.4 mg/dL) in comparison to the control group with a median of 0.4 mg/dL (range 0.3–1.0 mg/dL). Further differences were observed for the lactate levels in blood with a median of 1.0 mmol/l (range 0.4–16) and 1.4 mmol/l (range 1.0–2.4) for the study group and the control group, respectively.

There are several limitations to this study. The series is small so that statistical tests have to be read with caution. Also, collection of material for assessment did not follow a strict protocol. On the other hand, the study provides valid information regarding the exposure of patients with plasticizers. A strength of the study is the simultaneous quantification of the internal exposure of DEHP and its primary metabolite MEHP in the patients and the joint survey of chemical exposure and clinical parameters.

## Conclusions

The results of the study clearly demonstrate that ECMO patients received a considerable high internal exposure to DEHP, which is the still most prominent plasticizer in medical devices up to date. Moreover, the results indicate that the high DEHP exposure may contribute to the elevated blood levels of bilirubin in the patients. Due to the suspected toxicological potential of DEHP, its effects in patients and use in medical devices should be further evaluated, in particular for ICU patients with long-term medical treatments like ECMO therapy.

## Supporting information

**S1 Dataset. Dataset_ECMO_DEHP.**
(CSV)

## Acknowledgments

The authors are grateful to all patients participating in the study and to the staff of the intensive care unit of the Saarland University Medical Center.

## Author Contributions

**Conceptualization:** Franziska Kaestner, Hans Drexler, Philipp M. Lepper, Thomas Göen.

**Data curation:** Franziska Kaestner, Frederik Seiler, Daniel Rapp, Carlos Metz.

**Formal analysis:** Franziska Kaestner, Daniel Rapp, Thomas Göen.

**Investigation:** Franziska Kaestner, Frederik Seiler, Elisabeth Eckert, Johannes Müller, Thomas Göen.

**Methodology:** Franziska Kaestner, Johannes Müller, Thomas Göen.

**Project administration:** Franziska Kaestner, Hans Drexler, Philipp M. Lepper, Thomas Göen.

**Resources:** Franziska Kaestner, Hans Drexler, Thomas Göen.

**Supervision:** Franziska Kaestner, Robert Bals, Hans Drexler, Philipp M. Lepper, Thomas Göen.

**Validation:** Franziska Kaestner, Elisabeth Eckert, Thomas Göen.

**Visualization:** Franziska Kaestner, Thomas Göen.

**Writing – original draft:** Franziska Kaestner, Daniel Rapp, Elisabeth Eckert, Johannes Müller, Thomas Göen.

**Writing – review & editing:** Franziska Kaestner, Elisabeth Eckert, Philipp M. Lepper, Thomas Göen.

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
