## [Decision Letter · Decision Letter 0]

25 Sep 2019

PONE-D-19-22817

Internal exposure to di(2-ethylhexyl)phthalate (DEHP) and its metabolite MEHP during extracorporeal membrane oxygenation (ECMO) therapy

PLOS ONE

Dear Prof. Dr. med. Lepper,

Thank you for submitting your manuscript to PLOS ONE. After careful consideration, we feel that it has merit but does not fully meet PLOS ONE’s publication criteria as it currently stands. Therefore, we invite you to submit a revised version of the manuscript that addresses the points raised during the review process. 

We would appreciate receiving your revised manuscript by Nov 09 2019 11:59PM. To enhance the reproducibility of your results, we recommend that if applicable you deposit your laboratory protocols in protocols.io, where a protocol can be assigned its own identifier (DOI) such that it can be cited independently in the future. For instructions see: http://journals.plos.org/plosone/s/submission-guidelines#loc-laboratory-protocols

We look forward to receiving your revised manuscript.

Kind regards,

Jaymie Meliker, Ph.D.

Academic Editor

PLOS ONE

Journal Requirements:

2. Please provide additional details regarding participant consent. In the ethics statement in the Methods and online submission information, please ensure that you have specified (1) whether consent was informed and (2) what type you obtained (for instance, written or verbal). If your study included minors, state whether you obtained consent from parents or guardians. If the need for consent was waived by the ethics committee, please include this information.

3. Please include your tables as part of your main manuscript and remove the individual files. Please note that supplementary tables (should remain/ be uploaded) as separate "supporting information" files

I have read the journals policy and the authors of this manuscript have the following competing interests: FK received travelling fees from Getinge Deutschland GmbH; PL received travelling fees from Getinge Deutschland GmbH, and speaker fees from Mitsubishi Pharma, and CSL Behring. The other authors declared no conflict of interest.

Reviewers' comments:

Reviewer's Responses to Questions

**Comments to the Author**

1. Is the manuscript technically sound, and do the data support the conclusions?

Reviewer #1: Yes

2. Has the statistical analysis been performed appropriately and rigorously? 

Reviewer #1: Yes

3. Have the authors made all data underlying the findings in their manuscript fully available?

Reviewer #1: Yes

4. Is the manuscript presented in an intelligible fashion and written in standard English?

Reviewer #1: Yes

5. Review Comments to the Author

Reviewer #1: Interesting paper on exposure to DEHP during ECMO and other intense Medical treatment.

Specific comments:

1. Title: the term Internal exposure is strange, why not Intravenous exposure, as this is what you are measuring. Or simply Exposure.

2. I miss in the article any reference to the ongoing regulatory developments, with more stringent requirements for the use of DEHP, or plasticizers in general. This must be added.

3. Page 6, line 166: not sure if accuracy is the right word here, do you mean recovery?

4. Results, page 8: lots of data points for DEHP and MEHP are shown, but several questions remain: was there a correlation between days on ECMO and the levels determined. Was there a correlation between number of red cell transfusions and exposure. In figure 2 and 3: probably there are patients with more than two cannulas and more than one membrane; worthwhile to give this information. Are these patients also having more transfusions and other Medical treatment with PVC components?

5. In table 2 it would be of value to have also the DEHP and MEHP values in the blood added.

6. Discussion, page 10, line 269: maybe better to formulate differently, high values are found to be correlated with longer and more intense ECMO (several canulas, membranes), but not in all patients.

7. Discussion, page 11 line 294 ctd. and page 12 line 317 ctd. seems to be in contrast. Line 294 is 'could be' and in line 317 it is a hypothesis. In my opinion the main reason for higher DEHP in patients without urine output is inhibited elimination, and perhaps more intense treatment plays an additional role, according to the hypothesis.

8. Discussion, page 11, line 300: please add that cell saver treatment includes a washing step.

9. Discussion, page 11, line 314 ctc: according to the individual points in the graphs, it looks like low DEHP is also low MEHP, can you add a remark on this point to the discussion in the part where you discuss MEHP proportion.

10. Introduction: although it is true that DEHP is (re)classified as carcinogenic, it has to be clear that the values needed for this effect are far above those for the endocrine disruptive effects.

6. PLOS authors have the option to publish the peer review history of their article (what does this mean?). If published, this will include your full peer review and any attached files.

Reviewer #1: No

---

## [Author Response · Author response to Decision Letter 0]

22 Oct 2019

Reviewer #1: Interesting paper on exposure to DEHP during ECMO and other intense Medical treatment.

We thank the reviewer for his critical appraisal of our paper. We would like to answer the raised issues as follows: 

Specific comments:

1. Title: the term Internal exposure is strange, why not Intravenous exposure, as this is what you are measuring. Or simply Exposure.

We fully agree with the reviewer and hence change the title to simply „Exposure of…“

2. I miss in the article any reference to the ongoing regulatory developments, with more stringent requirements for the use of DEHP, or plasticizers in general. This must be added.

We agree and add the following paragraph in the manuscript in line 77-82: Additionally, DEHP in much higher values is classified as a carcinogen for which a non-genotoxic mode of action is predominant (8). The application of several phthalates is regulated in the European Union and DEHP and several other phthalates are banned for the manufacturing of children's toys (9, 10). In contrast, a restriction of DEHP use in medical devices does not exist. Nevertheless, the European Scientific Committee on Emerging and Newly-Identified Health Risks recommended a substitution of DEHP for medical devices, whenever possible (11)

3. Page 6, line 166: not sure if accuracy is the right word here, do you mean recovery?

As we fell, the reviewers suggestion gives more precision, we now write e.g. an accuracy (recovery) of 100–109 % for both analytes.

4. Results, page 8: lots of data points for DEHP and MEHP are shown, but several questions remain: was there a correlation between days on ECMO and the levels determined. Was there a correlation between number of red cell transfusions and exposure. In figure 2 and 3: probably there are patients with more than two cannulas and more than one membrane; worthwhile to give this information. Are these patients also having more transfusions and other Medical treatment with PVC components?

We agree with the reviewer and changed the manuscript in multiple aspects: 

Line 221 – 223: The decrease of MEHP after seven days of ECMO in one patient may be due to a limited metabolism but is not verified. The association between duration of ECMO therapy and DEHP or MEHP was tested using Spearman’s correlation coefficient. However, the correlation did not reach statistical significance (r = 0.16, P = 0.53).

Line 279 -290: In the present study high values of DEHP and its metabolites are found to be correlated with longer and more intense ECMO (several cannulas, membranes), but not in all patients. We observed a trend of increased DEHP levels after initiation of ECMO therapy. Furthermore, we observed that DEHP and its metabolites were increased when a more intense ECMO treatment (as measured by number of cannulas or number of membranes) was applied. However, there was no positive correlation between duration of ECMO therapy and DEHP levels. This might be due to a rapid increase of DEHP after initiation of the therapy and a steady state during the rest of the treatment. Other factors like the individual metabolism might also have a strong impact on the DEHP-levels. More studies with a larger sample sizes and a more frequent measurement of DEHP-levels are needed in order to further analyse this question.

Line 240 – 242: Correlation between DEHP and applied packed cells showed significance (spearman`s correlation coefficient r = 0.59, p<0.01).

And finally line 326 – 332: We observed a statistically significant correlation between DEHP-levels and number of packed cells given. This could be due to the critical condition of patients who received a high number of packed cells. Those critical patients might also have received a more intense ECMO therapy. On the other hand, we did not measure the DEHP-levels in the blood products. Former studies indicated a high contamination of the blood products with DEHP, which may have contributed to the DEHP-blood levels in those patients (5, 36).

5. In table 2 it would be of value to have also the DEHP and MEHP values in the blood added.

We agree and add these values. 

6. Discussion, page 10, line 269: maybe better to formulate differently, high values are found to be correlated with longer and more intense ECMO (several canulas, membranes), but not in all patients.

We changed this according to the reviewers suggestion (line 279 ff). 

7. Discussion, page 11 line 294 ctd. and page 12 line 317 ctd. seems to be in contrast. Line 294 is 'could be' and in line 317 it is a hypothesis. In my opinion the main reason for higher DEHP in patients without urine output is inhibited elimination, and perhaps more intense treatment plays an additional role, according to the hypothesis.

We thank the reviewer for this suggestion and we change the manuscript accordingly. 

8. Discussion, page 11, line 300: please add that cell saver treatment includes a washing step.

Done. 

9. Discussion, page 11, line 314 ctc: according to the individual points in the graphs, it looks like low DEHP is also low MEHP, can you add a remark on this point to the discussion in the part where you discuss MEHP proportion.

We agree with the reviewer and add in line 353 ff.: „DEHP and MEHP showed similar dependence, but the MEHP to DEHP ratio was not constant.“

10. Introduction: although it is true that DEHP is (re)classified as carcinogenic, it has to be clear that the values needed for this effect are far above those for the endocrine disruptive effects.

The reviewer is right in saying much higher levels are needed for carcinogenic effects, we thus state in line 77: „Additionally, DEHP in much higher values is classified as a carcinogen for which a non-genotoxic mode of action is predominant.“

---

## [Editor Report · Decision Letter 1]

25 Oct 2019

Exposure of patients to di(2-ethylhexy)phthalate (DEHP) and its metabolite MEHP during extracorporeal membrane oxygenation (ECMO) therapy

PONE-D-19-22817R1

Dear Dr. Lepper,

We are pleased to inform you that your manuscript has been judged scientifically suitable for publication and will be formally accepted for publication once it complies with all outstanding technical requirements.

With kind regards,

Jaymie Meliker, Ph.D.

Academic Editor

PLOS ONE
---

## [Editor Report · Acceptance letter]

23 Jan 2020

PONE-D-19-22817R1 

Exposure of patients to di(2-ethylhexy)phthalate (DEHP) and its metabolite MEHP during extracorporeal membrane oxygenation (ECMO) therapy 

Dear Dr. Lepper:

I am pleased to inform you that your manuscript has been deemed suitable for publication in PLOS ONE. Congratulations! Your manuscript is now with our production department. 

With kind regards,

on behalf of

Dr. Jaymie Meliker 

Academic Editor

PLOS ONE